# Drought Trend Analysis Based on the Standardized Precipitation–Evapotranspiration Index Using NASA's Earth Exchange Global Daily Downscaled Projections, High Spatial Resolution Coupled Model Intercomparison Project Phase 5 Projections, and Assessment of Potential Impacts on China's Crop Yield in the 21st Century

**Xiaolin Guo [1], Yuan Yang [1], Zhansheng Li [1,*], Liangzhi You [2], Chao Zeng [3], Jing Cao [4,*] and Yang Hong [5]**

1   State Key Laboratory of Hydroscience and Engineering, Department of Hydraulic Engineering, Tsinghua University, Beijing 100084, China; guoxl15@mails.tsinghua.edu.cn (X.G.); yangyuan15@mails.tsinghua.edu.cn (Y.Y.)
2   International Food Policy Research Institute, Washington, DC 20006, USA; l.you@cgiar.org
3   School of Resource and Environmental Sciences, Wuhan University, Wuhan 430000, China; zengchaozc@hotmail.com
4   School of Economics and Management, and Hang Lung Center for Real Estate, Tsinghua University, Beijing 100084, China
5   School of Civil engineering and Environmental Science, University of Oklahoma, Norman, OK 73019, USA; yanghong@ou.edu
*   Correspondence: lizs1985@tsinghua.edu.cn (Z.L.); caojing@sem.tsinghua.edu.cn (J.C.); Tel.: +86-10-62787394 (Z.L.)

**Abstract:** Drought is among the costliest natural disasters on both ecosystems and agroeconomics in China. However, most previous studies have used coarse resolution data or simply stopped short of investigating drought projection and its impact on crop yield. Motivated by the newly released higher-resolution climate projection dataset and the crucial need to assess the impact of climate change on agricultural production, the overarching goal of this study was to systematically and comprehensively predict future droughts at unprecedented resolutions over China as a whole. rather than region-specific projections, and then to further investigate its impact on crop yield by innovatively using a soil water deficit drought index. Methodologically, the drought projections were quantified from very high resolution climate data and further predicted impacts on crop yield over China using the standardized precipitation–evapotranspiration index (SPEI) at a relatively high (25 km) spatial resolution from NASA's Earth Exchange Global Daily Downscaled Projections (NEX-GDDP). The results showed that (1) overall, China is projected to experience a significant decrease in SPEI (−0.15/decade under RCP (representative concentration pathway) 4.5; −0.14/decade under RCP8.5). Seasonally, the decreasing rate of SPEI is projected to be largest in winter (−0.2/decade and −0.31/decade) and the least in summer (−0.08/decade and −0.10/decade) under respective RCPs. (2) Regionally, winter/spring will get drier, especially at high latitudes/altitudes (North China and Tibetan plateau), and summer/autumn will get wetter in southern China. (3) Both the frequency and duration for medium and severe drought are projected to decrease, while extreme drought, particularly in high latitudes/altitudes, is projected to increase. (4) The percentage of the potential crop production affected by drought would increase to 36% (47%) by 2100 under RCP4.5 (RCP8.5). Especially, the ratio impacted by extreme drought is projected to increase over time and with much worse magnitude under RCP8.5; thus, adaptive crop policies are expected to address such a risk.

**Keywords:** drought trend; SPEI; CMIP5 projection; crop yield; China

## 1. Introduction

Under climate change and warming temperature, the water cycle (including precipitation, evaporation, soil moisture and runoff, etc.) is consequently accelerating, increasingly causing extreme hydrometeorological or hydroclimatic events (such as floods and droughts) across the world [1–4]. The special report by the Intergovernmental Panel on Climate Change (IPCC SREX, 2012) also points out that more frequent and severe floods and droughts will likely continue to increase in the coming decades. Particularly, droughts have an immensely adverse impact on water resources and crop yields. As the most populous country in the world, China is still relying largely on its domestic agriculture despite the development of industry. Thus, China is very sensitive to climate change and related natural disasters. Among all these disasters, drought is the most severe one in China, causing massive losses in agricultural production. Nationally, drought has caused around 26 million tons of grain production annually in the last decade, which is about 5.2% of its total grain production [5].

Drought definitions vary and can be classified as meteorological drought, hydrological drought, agricultural drought, and socio-economic drought [6]. Accordingly, different drought indices are used to quantify drought impacts such as the Palmer drought severity index (PDSI) and the standardized precipitation index (SPI) for meteorological droughts [7–9]. Recently, the standardized precipitation–evapotranspiration index (SPEI; Vicente-Serrano et al. 2010) has been widely applied in soil water deficit conditions. Compared to other indices, the SPEI can better reflect the impacts of drought on agriculture, for it combines precipitation and potential evapotranspiration and has been increasingly used; however, such SPEI-based research on the impact of drought on maize yield has only been conducted in the plains of North China [6,7].

The historical spatiotemporal changes in wetting and drying areas over China have been investigated in previous research [7,10–13]. Recently, a series of severe and extensive droughts in both 2006 and from 2009 to 2011 had swept across southwest China, resulting in tremendous economic losses and disruption of society [14]. As calculated, the economic losses due to drought could reach up to 50% of the total losses from hydroclimatic disasters [5]. Therefore, there is a critical need to better predict droughts caused by climate change in order to help the government develop effective strategies in not just responding to agricultural loss but also to socioeconomical crisis.

Predicting future drought trends has received much attention in recent years, which is important for long-term water use planning and developing adaptation strategies to reduce climate risk [15–19]. However, most current work focuses on the analysis of specific regions (e.g., southwest China [20], eastern China [21], northwest China [22], and Loess Plateau [23]) rather than the whole China. Therefore, it is necessary to systematically and comprehensively predict drought trends nationwide. Moreover, future changes in drought have been analyzed under different representative concentration pathway (RCP) scenarios under the Coupled Model Intercomparison Project Phase 5 (CMIP5); and the reliability of CMIP5 to project the spatial-temporal variations of drought in China have been assessed and validated by several studies [14,24–26], but in general, they have been at coarse spatial resolutions of 0.5°–2°. Although these studies have suggested that both frequency and severity of drought would continuously increase with global warming, distinctive contradictions exist among these studies [27–32]. Significant discrepancies in drought trend predictions are caused by many reasons, like the uncertainties associated with different general circulation models (GCMs) and the magnitude of temperature increase, the choice of drought indices, and the methods to calculate drought characteristics [24,33–35].

Motivated by the higher-resolution NASA Earth Exchange Dataset and the crucial need to assess the impact of climate change on agricultural production, the overarching goal of this study is to systematically and comprehensively predict future droughts over China and to further investigate

its impact on crop yields. There are several distinctions and innovations compared to previous studies. Specifically, this study first predicts droughts over China as a whole rather than by region, thus providing a nationwide picture of droughts and their potential impact on crop yields. Second, considering the great heterogeneity of the topography and climates of China, this study aims to improve on detailed temporal–spatial drought predictions by using the relatively highest-resolution dataset possible, released by NASA Earth Exchange Global Daily Downscaled Projections (NEX-GDDP), at 25 km resolution and daily scales. Third, this study goes beyond drought prediction and extends its impact on crop yields by linking the soil-water deficit index (SPEI) with farmland potential yield data at 1 km resolution, including rice, maize, wheat, soybean, and sweet potato, which accounts for over 95% of the national total food output [36].

Utilizing the higher-resolution dataset released by NEX-GDDP, this paper presents projected drought changes defined by SPEI per year and per season on a decadal scale by using the multimodel ensemble (MME) of 10 downscaled CMIP5 model projections, with $25 \times 25$ km resolution, in the 21st century over China. First the quality of CMIP5 and MME results are evaluating reflecting the temporal–spatial variability of the monthly temperature and precipitation between 1979 and 2000. Then, the future drought spatiotemporal variability is calculated based on SPEI over China and 8 climate regions under different RCP scenarios (RCP4.5 and RCP8.5) in the 21st century. RCP4.5 and RCP8.5 are two radiative forcing pathways equal to 4.5 W/m$^{-2}$ (equivalent to 650 ppm $CO_2$ concentration) and 8.5 W/m$^{-2}$ (equivalent to 1370 ppm $CO_2$ concentration) total emissions released by 2100, respectively, and these two scenarios are widely used to analyze climate change [4,11,14,15,24–26]. Finally, the frequency and duration of different drought classes as well as the potential impact on crop production are investigated and discussed.

This paper is organized as follows. Section 2 describe the datasets, followed by the general methods, and climate zones of the study area are discussed in Section 3. Section 4 presents the results and analysis, followed by the summary and conclusion in Section 5.

## 2. Datasets: CMIP5 Simulations and Forcing Data

Based on global climate model simulations from CMIP5 [33], the NEX-GDDP project provides downscaled global climate scenarios at a spatial resolution of 0.25° (about $25 \times 25$ km) [34], including 21 GCM modes and across two greenhouse gas emission scenarios (RCP4.5, RCP8.5). The variables in the NEX-GDDP project include daily precipitation (Pre), maximum daily temperature (Tasmax), and minimum daily temperature (Tasmin). The downscaled method used in the NEX-GDDP dataset and widely used in climate change simulations is the bias-corrected spatial disaggregation (BCSD) method [37–39]. In BCSD, the global observed data used as reference are from the Global Meteorological Forcing Dataset for Land Surface Modeling, which was produced by the Terrestrial Hydrology Research Group at Princeton University [40]. More detailed information can be found in the paper of Thrasher et al. (2012) [31] and on the official website (https://cds.nccs.nasa.gov/nex-gddp/).

NEX-GDDP has proved to be able to give more precise and reliable projections compared to CMIP5 GCMs [41,42]. Ten representative models that have high original spatial resolutions and are from different countries and research groups were chosen in this study. All of these models have been applied in climate change research. The institution, country, and original spatial resolution are listed in Table 1. RCP4.5 and RCP8.5 are two radiative forcing pathways equal to 4.5 W/m$^{-2}$ (equivalent to 650 ppm $CO_2$ concentration) and 8.5 W/m$^{-2}$ (equivalent to 1370 ppm $CO_2$ concentration) total emissions released by 2100, respectively. These two scenarios are widely used to analyze climate change, as they represent the expected and the worst scenarios in the future [43]. Based on 10 CMIP5 models that had available daily outputs, including Tasmin, Tasmax, and Pre from two scenarios, the MME was calculated by using unweighted, average monthly Pre, Tasmin, and Tasmax values for the following SPEI calculation.

**Table 1.** Information on the 10 climate models used in the present analysis.

| Institution | Model | Spatial Resolution (Lon×Lat, Degree) | Country |
|---|---|---|---|
| Centre National de Recherches Météorologiques/Centre Européen de Recherche et Formation Avancée en Calcul Scientifique (CNRM–CERFACS) | CNRM-CM5 | $1.40° \times 1.40$ | French |
| Atmosphere and Ocean Research Institute (The University of Tokyo), National Institute for Environmental Studies, and Japan Agency for Marine-Earth Science and Technology (MIROC) | MIROC5 | $1.40° \times 1.40°$ | Japan |
| National Science Foundation, Department of Energy, National Center for Atmospheric Research | CESM-BGC | $0.94° \times 1.25°$ | USA |
| Commonwealth Scientific and Industrial Research Organization (CSIRO) and Bureau of Meteorology (BOM), Australia | ACCESS1.0 | $1.25° \times 1.875°$ | Australia |
| National Center For Atmospheric Research (NCAR) | CCSM4 | $0.94° \times 1.25°$ | USA |
| Commonwealth Scientific and Industrial Research Organization in collaboration with the Queensland Climate Change Centre of Excellence(CSIRO-QCCCE), Australia | CSIRO-Mk3.6.0 | $1.875° \times 1.875°$ | Australia |
| Institute for Numerical Mathematics(INM), Russia | INM-CM4 | $2.0° \times 1.5°$ | Russia |
| Institute Pierre-Simon Laplace (IPSL), France | IPSL-CM5A-MR | $1.267° \times 3.750°$ | France |
| Max Planck Institute for Meteorology(MPI-M),Germany | MPI-ESM-MR | $1.875 \times 1.875$ | Germany |
| Meteorological Research Institute(MRI), Japan | MRI-CGCM3 | $1.125 \times 1.125$ | Japan |

ITPCAS data (China Meteorological Forcing Dataset, which was developed by the Data Assimilation and Modeling Center for Tibetan Multispheres, Institute of Tibetan Plateau Research, Chinese Academy of Science (ITPCAS)) from 1979–2000 were used to assess and validate the downscaled temperature and precipitation projections from the NEX-GDDP CMIP5 model. The ITPCAS forcing dataset had a 0.1° spatial and 3 h temporal resolutions. The accuracy of the dataset has been evaluated, and the previous research shows that meteorological parameters from this dataset have less root-mean-square errors ($RMSE = \sqrt{\sum_1^n \theta_i^2}$; we assume that we have n samples, and $\theta$ represents model error; RMSE has been used as a standard statistical metric to measure model performance [44]) and higher correlation coefficients (CC) with ground measurements over the other dataset. Besides, this dataset has already been used in land–atmospheric interactions and force land–surface models in China [45–47]. More information is detailed in He et al. (2010) [48].

## 3. General Method and Study Area

### 3.1. The General Method: Standardized Precipitation–Evapotranspiration Index and the Hargreaves–Samani Equation

Standardized precipitation–evapotranspiration index calculations have been carried out by Vicente-Serrano et al. in 2010:

$$SPEI = W - \frac{c_0 + c_1 W + c_2 W^2}{1 + d_1 W + d_2 W^2 + d_3 W^3}; F(x) = [1 + (\frac{\alpha}{x-y})^\beta]^{-1}$$

where *F(x)* is the probability distribution function of series *D*; and *α*, *β*, and *γ* are scale, shape, and origin parameters of the Pearson III distribution, respectively (Singh et al., 1993; Vicente-Serrano et al., 2010a).

For $P(D) \leq 0.5$, $W = \sqrt{-2lnP(D)}$, $P(D) = 1 - F(x)$; for $P(D) > 0.5$, $P(D)$ is replaced by $1 - P(D)$, and the sign of SPEI is reversed. The constants are: $C_0 = 2.515517$, $C_1 = 0.802853$, $C_2 = 0.010328$, $d_1 = 1.432788$, $d_2 = 0.189269$, and $d_3 = 0.001308$ [49]. SPEI is a multiscale drought index based on climatic data. It can describe water deficit effectively over multiple time scales, reflecting the lag relationship between different water resources, precipitation, and evapotranspiration [14]. Three-month SPEI data were employed in this study to characterize short duration droughts [50]. More details about SPEI can be obtained online (http://sac.csic.es/spei/index.html) and in previous papers [49,51–53]. SPEI is preferred compared to the standardized precipitation index (SPI) [54] since the former considers the effect of the atmospheric circumstances to the water balance by introducing evapotranspiration. Under global warming, the SPEI not only retains the multiscale characteristics of droughts [49] in SPI, but it also helps to account for the impact of temperature. Therefore, SPEI has been nominated as a substitute of SPI as well as Palmer drought severity index (Palmer, 1965) to evaluate climatic water imbalance and drought [54–57].

In order to evaluate water deficit, the temperature and potential evapotranspiration (PET) were originally calculated with the classic Thornthwaite method (where water deficit (D) = precipitation – PET) [49]. Compared to the Thornthwaite method, the Hargreaves–Samani (HS) equation (Hargreaves and Samani, 1985) is as follows:

$$\text{ET}_{\text{HS}} = 0.0135 * (\text{KT}) * (\text{T}_{\text{av}} + 17.8) * (\text{T}_{\text{max}} - \text{T}_{\text{min}})^2 * \text{Ra} * 0.408 * \text{d}$$

where $\text{ET}_{\text{HS}}$ is evapotranspiration in mm/dekad; KT is the empirical coefficient; $\text{T}_{\text{max}}, \text{T}_{\text{min}}$ are the dekadal mean, maximum, and minimum air temperatures in °C; Ra is extraterrestrial radiation (MJm$^{-2}$); and d is the number of days within the decade. The HS equation shows advantages in both accounting for atmospheric information and capturing part of the land surface properties (soil moisture, land surface) [58]. The HS equation has been evaluated across different climate regions [59–63] and also has been used in SPEI calculations in previous works [55]. In addition, the Food and Agriculture Organization of the United Nations (FAO) recommends to utilize the HS equation when meteorological data are lacking to compute the potential evapotranspiration through the Peaman–Montie equation. Therefore, in our work, SPEI was adopted with potential evapotranspiration estimated using the HS equation. A three-month timescale was chosen for SPEI in drought projection, which could be considered an agricultural drought [64,65]. The criterion of determining drought intensities by the SPEI value can be found in Table 2 [9].

**Table 2.** The standardized precipitation–evapotranspiration index (SPEI) categories based on the initial classification of SPEI values.

| Drought Classification | SPEI Value (Probability) |
| :---: | :---: |
| Extreme humid | SPEI ≥ 2.0 (2.3%) |
| Severe humid | 1.5 ≤ SPEI < 2.0(4.4%) |
| Moderate humid | 1.0 ≤ SPEI < 1.5(9.2%) |
| Normal | −1.0 < SPEI < 1.0(68.2%) |
| Moderate dry | −1.5 < SPEI ≤ −1.0(9.2%) |
| Severe dry | −2.0 < SPEI ≤ −1.5(4.4%) |
| Extreme dry | SPEI ≤ −2.0(2.3%) |

### 3.2. Mann–Kendall (MK) and Sen's Tests

Compared to other parametric tests (e.g., *t*-test), the MK test can better analyze abnormally distributed data, even with outliers and missing values [62]. The significance of precipitation, temperature, and streamflow are estimated by the MK test [66–70]. $Z_{mk}$ in the MK test is calculated as described in Mann (1945), where a value greater than zero indicates a positive correlation with time. Testing trends are analyzed at a specific significant level ($\alpha = 0.05$). After the trend direction is derived by the MK test, the magnitude is determined by "Sen's estimator" method [71].

The $Z_{mk}$ in the MK test is calculated as described in Mann (1945) [72,73]. Positive values of $Z_{mk}$ indicate increasing trends, while negative $Z_{mk}$ values imply decreasing trends. Testing trends are analyzed at a specific α significance level. When $|Z_{mk}| > Z_{1-\frac{\alpha}{2}}$, the null hypothesis will be rejected, and a significant trend exists in the time series. $Z_{1-\frac{\alpha}{2}}$ can be obtained from the standard normal distribution table. In this study, a significance level of α = 0.05 was used. At the 5% significance level, the null hypothesis of no trend will be rejected if $|Z_{mk}| > 1.96$.

### 3.3. Climate Regions of the Study Area

To apply the abovementioned general methods and analyze the temporal–spatial pattern of drought changes from a climate zonal perspective, this study divided China into eight subregions (Figure 1) according to the topography and climate features following the National Assessment Report of Climate Change (National Report Committee, 2011) [64]. Precipitation and temperature in China are highly uneven in space and time. Previous research has found that mean precipitation decreases from southeast to northwest and has varied from 15 to over 2700 mm; and the mean temperature gradually decreases from south to north and has varied from −12 to 25 °C annually during 1961–2013 [10,11].

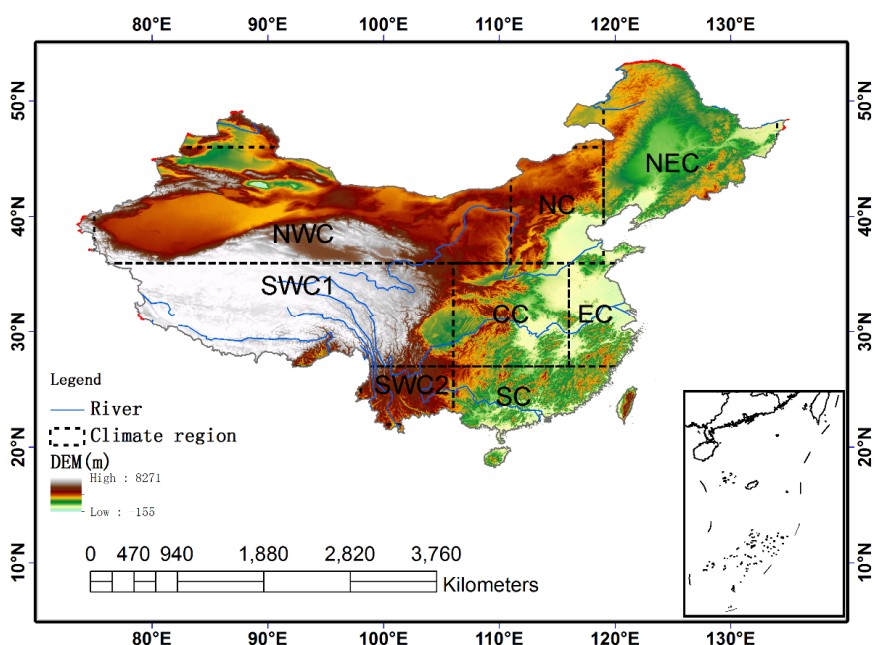

**Figure 1.** Eight subregions in China. NEC: Northeast China; NC: North China; EC: East China; CC: Central China; SC: South China; SWC1/TP: Tibetan Plateau; SWC2/SWC: Southwest China; and NWC: Northwest China.

## 4. Simulation Results and Discussion

### 4.1. Validation of CMIP5 and MME Temperature and Precipitation Simulations

The monthly precipitation and temperature were first evaluated from NEX-GDDP CMIP5 simulations since SPEI calculations were highly dependent on them. In order to validate the NEX-GDDP CMIP5 projections, a correlation coefficient and mean error (ME)/bias were employed to indicate the temporal agreement and magnitude of difference between monthly downscaled CMIP5 model projections (including the MME dataset) and ITPCAS data from 1979–2000.

Heatmaps of correlation coefficient and ME for the Tasmin and Tasmax over eight climate regions are presented in Figure 2. There is great discrepancy in the performance of temperature simulations among different CMIP5 models. CNRM-CM5, ACCESS1-0, MRI-CGCM3, and MPI-ESM-MR performed better in capturing monthly temporal variations than other CMIP5 models, but there was no significant difference in ME among different CMIP5 models.

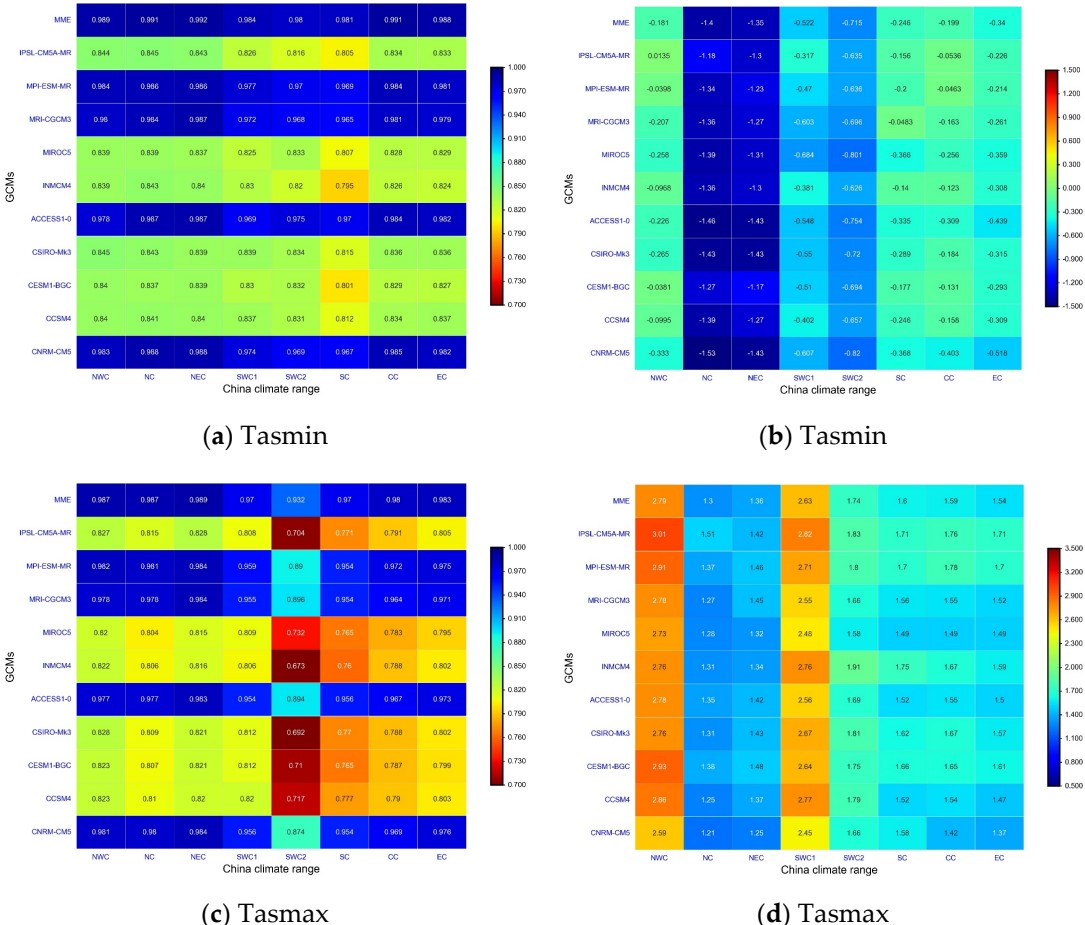

**Figure 2.** Correlation coefficient (**a**,**c**) and mean error (ME) (unit: C) (**b**,**d**) between monthly Tasmin (first row)/Tasmax (bottom row) from the Institute of Tibetan Plateau Research, Chinese Academy of Science (ITPCAS) and simulations from Coupled Model Intercomparison Project Phase 5 (CMIP5) models as well as the multimodel ensemble (MME) over eight climate regions from 1979–2000.

In general, the CC of Tasmin was lowest in South China (SC), and Southwest china (SWC2) was the region with the lowest CC for Tasmax. The ME heatmap (Figure 2b) shows that all ten models underestimated the Tasmin in China, with the largest discrepancy observed in North China (NC) and Northeast China (NEC). By contrast, Tasmax was overestimated by CMIP5 models, with the largest discrepancy observed in Northwest China (NWC) and the Tibetan Plateau (SWC1). Most importantly, the magnitude of discrepancy for Tasmax estimations (Figure 2d) was significantly larger than that of Tasmin (Figure 2b).

As presented in Figure 3, the performance of using GCMs to predict temperature was better than using GCMs to predict precipitation (Figure 2). The performance of the individual model for precipitation predictions on the regional scale was similar to that for temperature but with a stronger spatial variation (Figure 3). These ten models showed good performance over NC, NEC, SWC and SWC2 with respect to correlation coefficient compared to other regions (Figure 3a). In terms of the simulation error (i.e., bias) all models tended to overestimate precipitation in SWC1/SWC2 but underestimate precipitation in other regions, especially in NWC (Figure 3b).

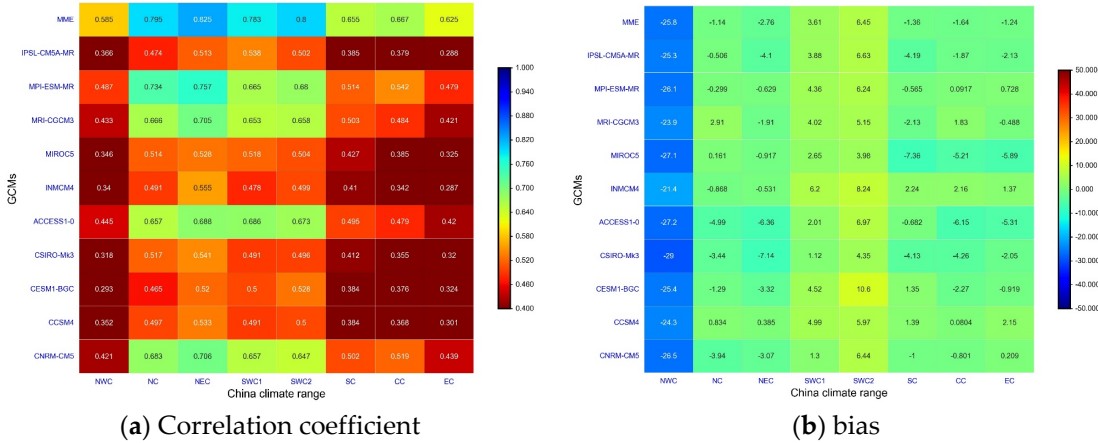

(**a**) Correlation coefficient                    (**b**) bias

**Figure 3.** Correlation coefficient (**a**) and bias (unit: %) (**b**) between ITPCAS monthly precipitation and simulations from the CMIP5 model as well as MME over eight climate regions from 1979–2000.

In terms of absolute magnitude, the performance (ME for temperature and bias for precipitation) of MME was similar to other models (Figures 2 and 3). However, MME could greatly increase the correlation coefficient compared to all individual GCM models for both temperature and precipitation predictions.

*4.2. Projected Change of SPEI_3 on a Decadal Scale*

For the two scenarios, SPEI_3 was projected to decrease 0.15 and 0.14 per decade under RCP4.5 and RCP8.5, respectively, in China, indicating that it will become drier in China in this century. Figure 4 shows the spatial distributions of SPEI_3 changes. The change of SPEI_3 under RCP4.5 (Figure 4a) shows that, except southern China (including SC, Central China, SWC, and East China (EC)) where drying or wetting trends were insignificant at the 5% significant level, all other regions including NC, NWC, and SWC1 were projected to become drier. In general, the SPEI_3 change rate was projected to increase from northwest to southeast, which implies a trend of becoming wetter in this direction.

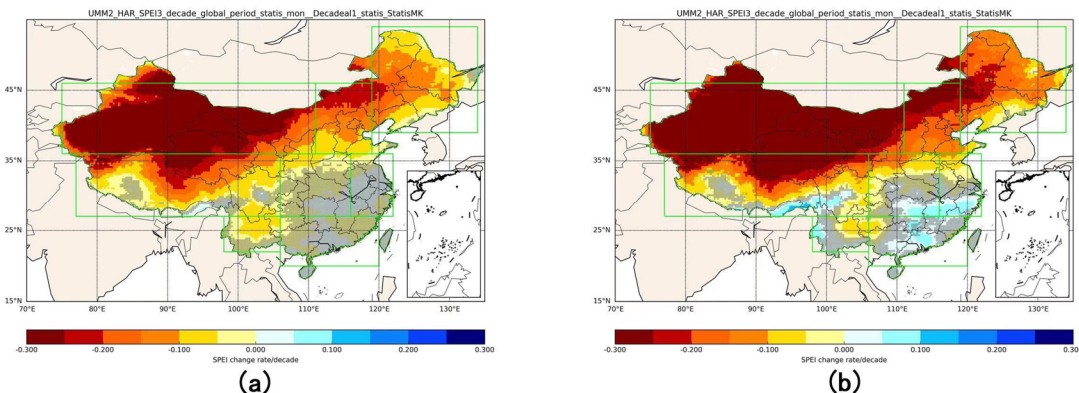

**Figure 4.** Spatial distribution of projected Sen's slope estimates for decadal SPEI_3 during the period 1950–2100 under scenarios RCP4.5 (**a**) and RCP8.5 (**b**) by using MME. The value in each pixel equals the slope of $\{SPEI_{1950–1960}, SPEI_{1960–1970}, \ldots, SPEI_{2000–2100}\}$. The cross-hatches represent that no statistically significant area existed, based on Mann–Kendall (MK) trends, in these pixels.

Under RCP8.5, areas in Southern China were predicted to get wetter compared to that under RCP4.5. Noticeably, areas where no significant wetting was projected under RCP4.5 had statistically significantly wetting under RCP8.5. In addition, the regions with insignificant drying on the border of SWC and the southern part of SWC1 under RCP4.5 also showed moderate increases in SPEI (i.e., they became wetter).

The change of SEPI on the regional scale is summarized in Table 3. High-latitude and high-altitude regions will get drier under both scenarios, although the drying rate of Southern China was obviously smaller compared to high-latitude and high-altitude regions. Except Central China and SWC under RCP4.5 showing a significant drying trend, there were no significant changes for dryness/wetness over EC, SC, and SWC under RCP8.5 on the regional scale.

**Table 3.** MK test and Theil-Sen's slope estimation for SPEI on a regional scale.

|  | RCP4.5 (SPEI/Decade) | RCP8.5 (SPEI/Decade) |
| --- | --- | --- |
| Northwest | * −0.285 | * −0.348 |
| North | * −0.161 | * −0.2299 |
| Northeast | * −0.068 | * −0.1048 |
| Tibetan Plateau | * −0.107 | * −0.124 |
| Southwest | * −0.037 | −0.0005 |
| South | −0.001 | 0.0035 |
| Central | * −0.017 | * −0.0367 |
| East | −0.003 | −0.014 |
| National | −0.150 | −0.140 |

* Significant trends > 5%. Negative values represent decreasing trends.

In the future, SPEI_3 changes show a pattern in which arid and semiarid regions (i.e., high-latitude and high-altitude regions) become significantly drier, and humid regions (i.e., over Southern China) would become wetter under RCP8.5. Although, results here are contradictory to previous findings that NWC and NC will get wetter [27,32,34]. This result is consistent with a well-known pattern, "drier regions are more likely to become drier, whereas wetter regions are more likely to become wetter" [22].

### 4.3. Change of Seasonal SPEI_3 on a Decadal Scale

Compared to the patterns on an annual scale, as shown in Figure 4, there were larger temporal–spatial variations in dry and wet conditions on the seasonal scale (Figure 5). In winter, the spatial patterns of SPEI changes were similar under the two scenarios, except a significantly larger drying rate was observed over Northern China, especially over NWC under RCP8.5 (Figure 5a,b).

In spring, the projected SPEI showed spatial patterns of a slight wetting condition over Central China and EC and drying conditions over other regions under RCP4.5. This spatial pattern intensified under RCP8.5. The transition zone between the dry regions and the wet regions was projected to experience significant drying. Meanwhile, Central China and EC regions, which were insignificantly wet under RCP4.5, were projected to get significantly wetter (Figure 5c,d).

In general, the changes in spatial patterns of SPEI in summer, autumn, and on an annual scale were similar (Figure 4), though more wet regions existed and were located all over China, except NWC and small patches in the northern NWC1. Similar to the changes in other seasons, this pattern will intensify under higher emissions. The SPEI changes in autumn were similar to those in summer (Figure 5e,f). The major difference was observed in Central China; different from the prediction that wet conditions will occur in Central China and its surroundings in summer, these regions were projected to get drier in autumn (Figure 5g,h).

Overall, it is projected that China will get drier in the four seasons. The seasonal decreasing rates of SPEI_3 were −0.18, −0.08, −0.11, and −0.2 per decade from spring to winder under RCP4.5. As for scenario RCP8.5, the rates were −0.27, −0.10, −0.14, and −0.31 per decade. Winter is projected to have the most serious decreasing rate, while summer has the least decreasing rate.

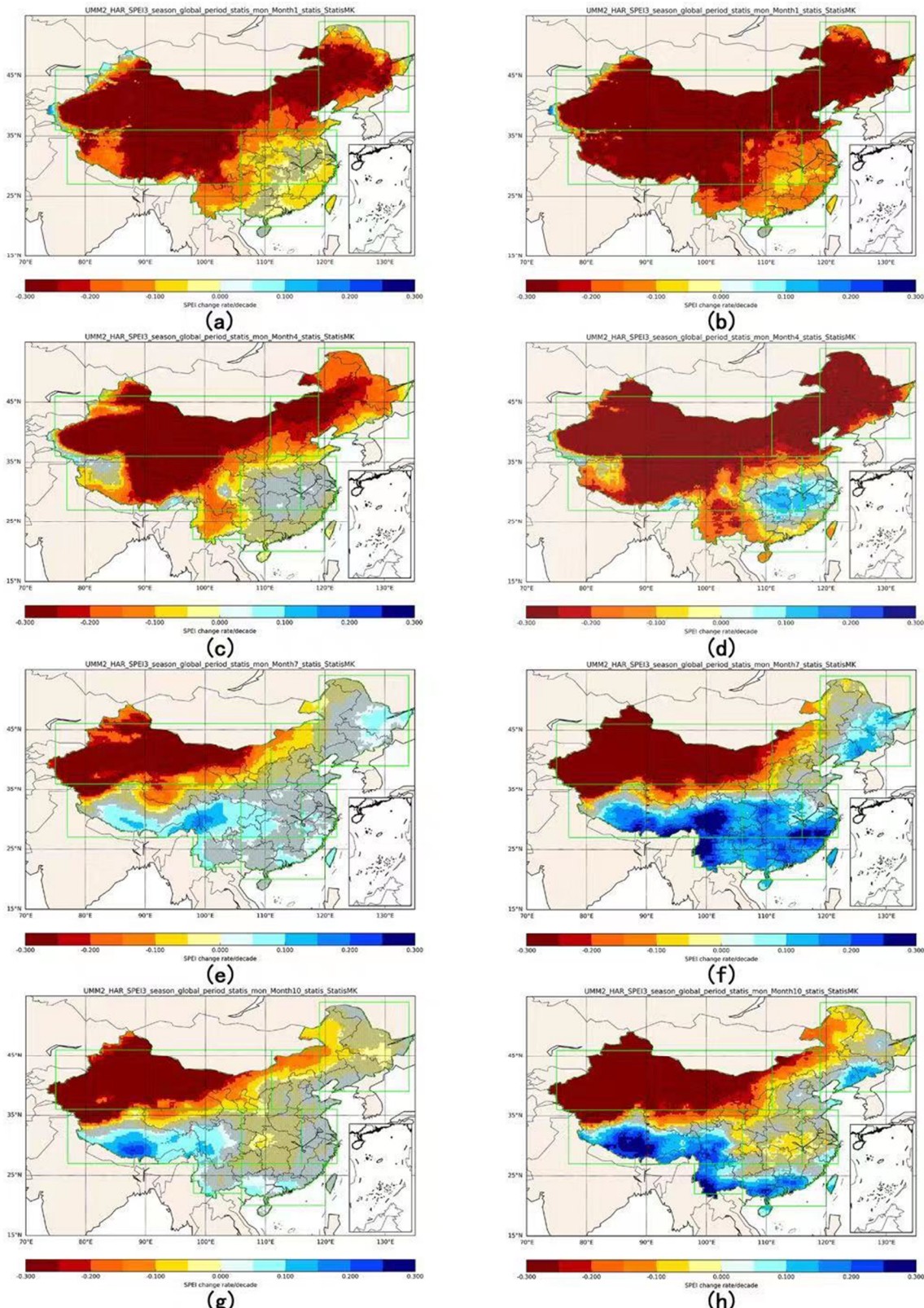

**Figure 5.** Spatial distribution of the projected four seasons using Sen's slope estimates for decadal SPEI_3 changes during the period 2000–2100 (top row is winter, followed by spring, summer, and autumn) by using MME. Note that the cross-hatches represent no statistical significance at the 5% level for Mann–Kendall trends in these pixels.

The heatmap in Figure 6 shows that all regions in China experienced significant drying in winter and spring. In addition, except NWC, NC, and Central China in autumn, the projection showed wetter conditions in China in summer and autumn. This wetting trend was more significant under RCP8.5. In the four seasons, the strongest drying trend occurred in NWC, while the least was observed in EC.

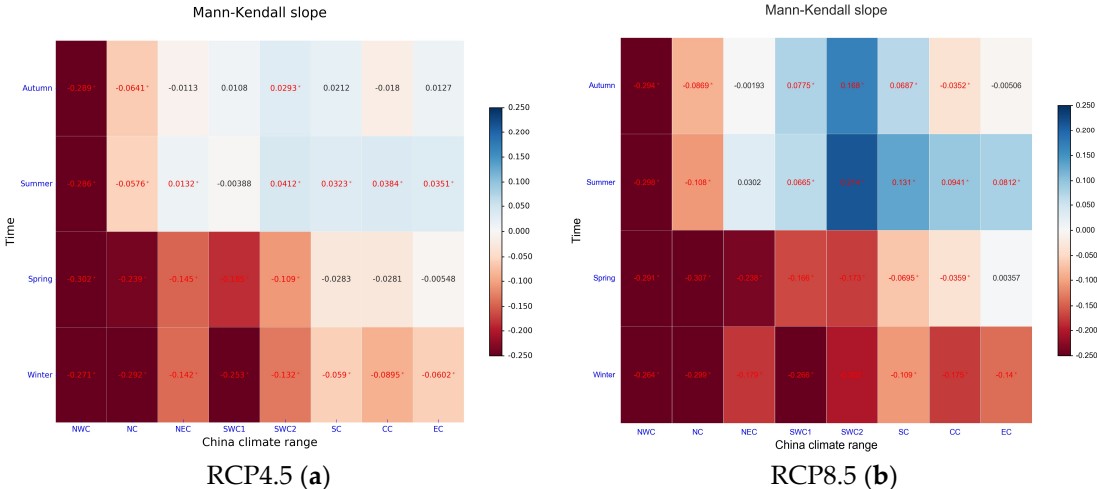

**Figure 6.** Heatmap of the projected four seasons based on Sen's slope estimates for decadal SPEI_3 changes during the period 2000–2100 over eight subregions, defined by Table 1, using MME under two scenarios. The slope with the red color and star symbol indicates that the Mann–Kendall trend is statistically significant at the 5% level.

In terms of the spatial pattern of dry/wet conditions, results in this work indicate wetter conditions in regions of SC and SWC, especially under RCP8.5. This prediction is contradictory to a previous conclusion that these regions are getting drier [27,37].

Moreover, the seasonal patterns presented in this work are also different from the conclusion drawn by Liu [14]. According to their study, there would be warm season dryness and cold season dryness in NC and SC, respectively. Instead, our results showed that summer and autumn is getting wetter, especially in SC. This difference might be caused by the different CMIP5 model data and the PET method in SPEI calculations used in our study.

SWC is projected to experience a medium level of drying both on the annual scale (Figure 4) and in winter and spring, and wetter conditions will be present in summer and autumn. This trend is also different from the works of Wang (2014) and Yang (2015). As their works suggest, SWC was projected to experience severe drought. In addition, drought changes over the Tibetan Plateau were similar to Southwest China, rather than the region with the most serious drier conditions in China. All of these discrepancies might be due to different drought indexes used in studies.

### 4.4. The Change of Drought Frequency and Duration on a Decadal Scale

The zone-averaged drought characteristics (i.e., drought duration and frequency) are presented in Figure 7. Different from the stable linear change of drought duration (Figure 7c,d), there were large temporal variations in the drought frequency (Figure 7a,b).

Figure 7 shows a decreasing or statically insignificant trend in drought frequency in these eight climate regions. Compared to drought frequency, an increasing trend of drought duration dominated in China. In addition, these results revealed that a similar pattern of decreasing frequency and increasing duration was projected across China under both RCP8.5 and RCP4.5 conditions, and this pattern had a stronger magnitude under RCP8.5.

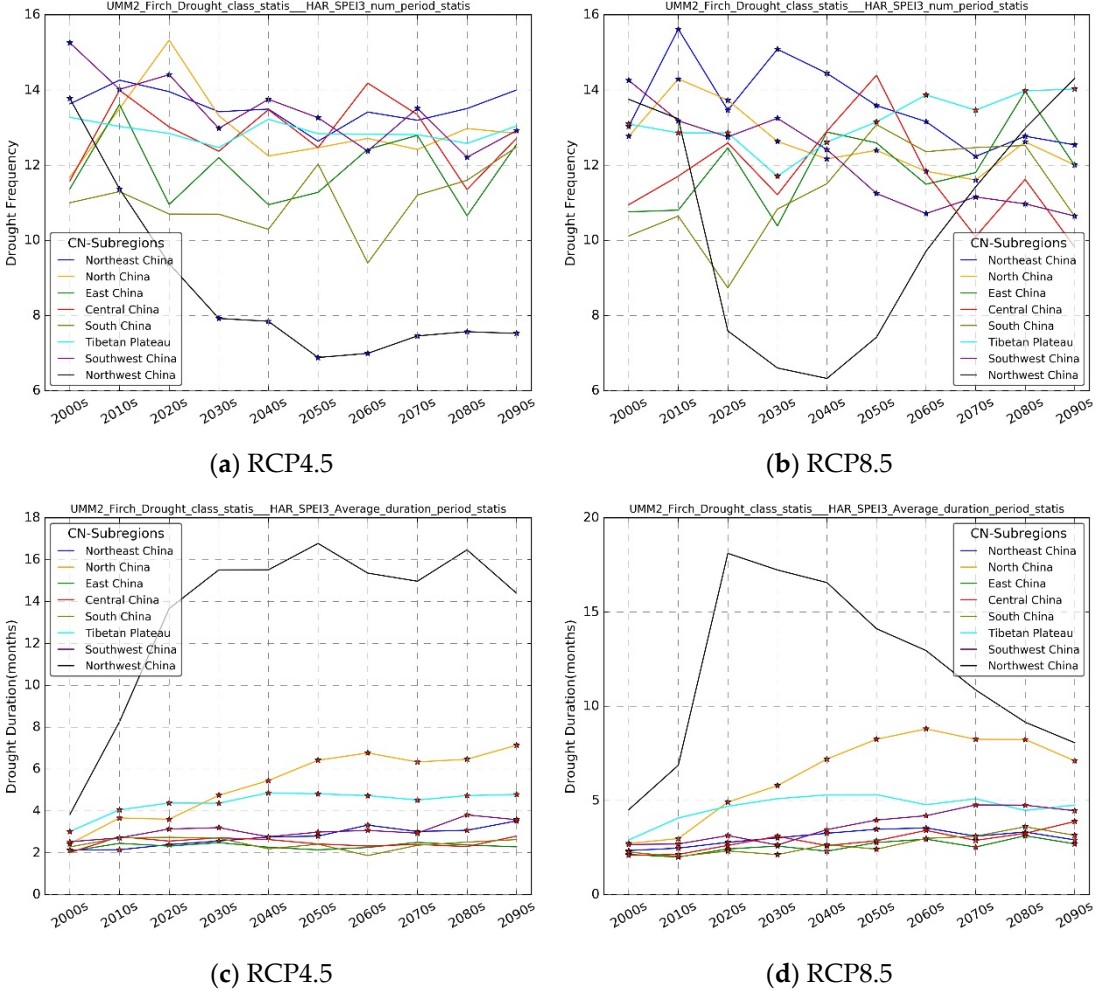

**Figure 7.** Temporal variation of regional drought frequency (top row: (**a**,**b**)) and drought duration (bottom row: (**c**,**d**)) on the decadal scale over eight climate regions. The red-filled star symbol represents positive trends, while the blue-filled star symbol represents negative trends.

Although the magnitude and change of NWC were the largest over China under two scenarios, the temporal variation in drought characteristics over NWC was remarkably different from that in other regions. Under RCP4.5, the frequency will decrease and the duration will increase in NWC, then both them will be kept stable. Under RCP8.5, drought frequency will increase stably after a dramatic decrease. Noticeably, the trend of duration change was opposite to that of frequency change. This complex temporal variation in NWC was possibly caused by the significant spatial heterogeneity over NWC. For moderate drought under RCP4.5, both frequency and duration would decrease in the 21st century (Figure 8a–d). A similar trend was observed for the severe drought.

As for moderate or severe drought, eight climate regions over China could be divided into two classes: high-latitude and high-elevation regions, which includes, respectively, NWC, NC, NEC and the Tibetan Plateau, and southern China as well as EC, Central China, SC, and SWC. In general, the drought frequency and duration in high-latitude and high-elevation areas were smaller than those in Southern China in the case of moderate and severe drought. In addition, both drought frequency and duration decreased significantly at the 5% significance level in high-latitude and high-elevation areas. The largest decline was predicted to occur in Northwest China.

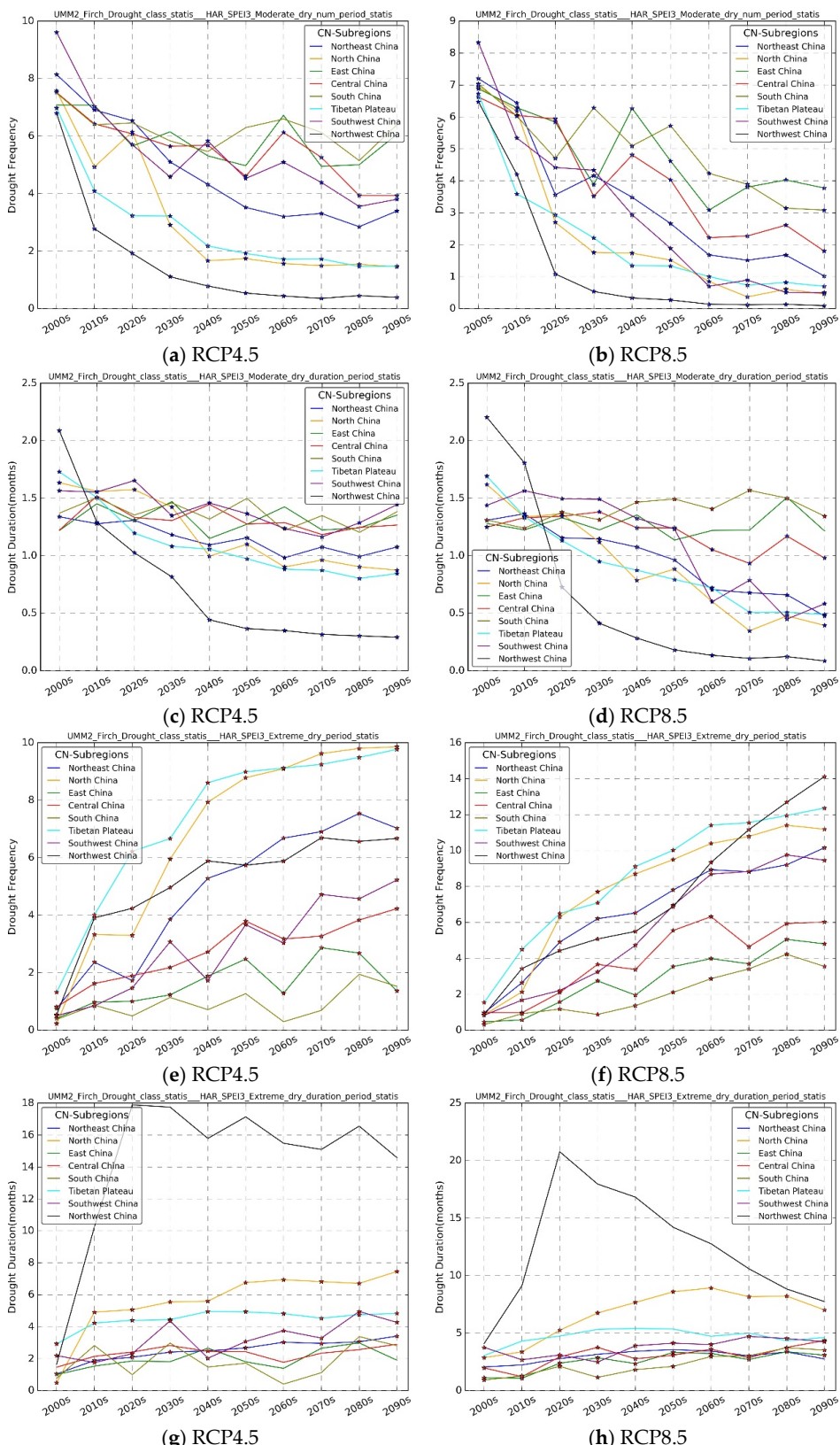

**Figure 8.** Temporal variation of regional drought frequency—moderate drought: (**a**,**b**); extreme dry: (**e**,**f**)—and drought duration—moderate drought: (**c**,**d**); extreme drought: (**g**,**h**)—on the decadal scale under two scenarios over eight climate regions. The red-filled star symbol on the line represents positive trends, while the blue-filled star symbol represents negative trends.

In the case of extreme drought, China will experience a significant increase in duration and frequency. In contrast to moderate drought, the duration and frequency was much larger in high-latitude and high-elevation areas compared to that in Southern China (Figure 8e–h). NC and the Tibetan Plateau had the largest frequencies. Although there was no significant trend for the change in duration in Northwest China, the magnitude of duration in Northwest China was the largest. This duration shortened in the order of North China, the Tibetan Plateau, and Southwest China. The shortest duration will be in South China.

Under RCP8.5, a similar, but intensified, pattern in frequency and duration changes was observed compared to that under RCP4.5. For moderate and severe drought, the lowest frequency and duration was projected to occur in NWC, while the highest was in SC and EC. Regions experiencing extreme drought were predicted to experience a higher frequency of drought. A significant decreasing trend was observed in NWC after the 2020s, which was different compared to other regions. However, the magnitude of duration was still remarkably larger than any other region. NC and SWC1 remain to be the regions with the highest frequency and the longest duration under RCP8.5.

Although previous work indicated that the frequency, strength, and duration of drought will increase in the future (IPCC SREX, 2012) [12,14,17,18], results here show that this trend is only observed for extreme drought. Instead, the frequency and duration are supposed to decline for moderate and severe drought.

### 4.5. Potential Risk for Crop Production Caused by Future Drought

Climate conditions are closely connected to natural resources and agricultural yield, and understanding how climate conditions will interact with society is vital to respond to potential risk. China has undergone great food demand because of the large population base and climate and land use changes. Recently, many researchers have focused on climate change and the corresponding agricultural loss [6,53,64], and they have even shown that three-month SPEI could effectively evaluate the effect of drought on crop yield [6,50]. Simulation of the potential yield under different drought scenarios were based on farmland potential yield data in 2010 (http://www.resdc.cn/DOI/doi.aspx?DOIid=43) and performed at a 1 km resolution. The main crops include rice, maize, wheat, soybean, and sweet potato, which represent over 95% of the national total food output [53,64].

The red line in Figure 9 shows the percentage of total potential agriculture production affected by drought. Significant, increasing trends in Figure 9a,b represent more loss of potential cropland production, and the total ratios will reach up to 37% (RCP4.5) and 48% (RCP8.5) by the end of this century. Figure 9 also represents the percentage of potential production loss caused by different levels of drought (middle, severe, and extreme drought), in which the relative ratio caused by severe and extreme drought notably increases and will reach up to 52% and 83% under RCP4.5 and RCP8.5 scenarios. One explanation of the loss is that the crop growth duration becomes shorter because of the increased temperature [53].

Our results provide more understanding of the impact of climate change on farmland potential yield and highlight the need for corresponding countermeasures. Additionally, research is needed on how planting structures, industrialization, and policies will influence agriculture and the economy.

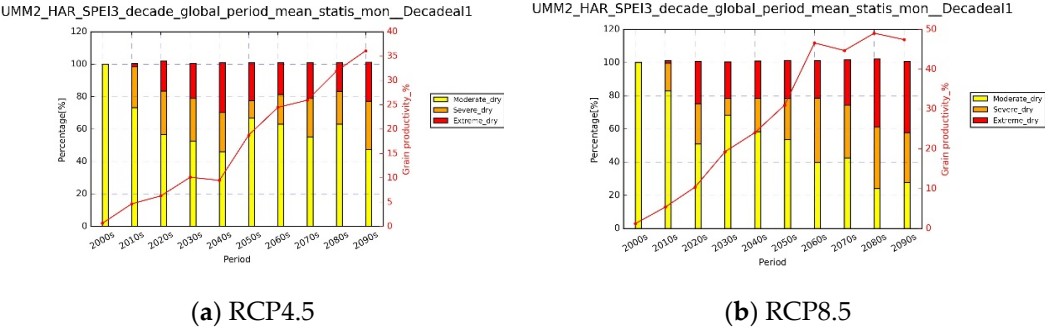

(**a**) RCP4.5          (**b**) RCP8.5

**Figure 9.** The simulation of farmland potential yield under two RCPs based on CMIP5: the red line (right axis) represents the percentage of potential total agriculture production affected by drought; the bar graph (left axis) represents the percentage of potential production loss caused by different levels of drought.

## 5. Conclusions

Based on ten CMIP5 models from the NEX-GDDP project, with a relatively high spatial resolution of 25 × 25 km, this work assessed CMIP5 projections with ITPCAS forcing data from 1979 to 2000 as the reference, and it projected the changes of dry/wet conditions over China by using SPEI with the Hargreaves–Samani equation during this century under RCP4.5 and RCP8.5. Since the data used in this research are available on a global scale, the methodology implemented in this study can be potentially applied in other drought-prone or water-scarce basins in the world to more specifically and efficiently cope with future drought impacts on crop yields.

In terms evaluating the NEX-GDDP CMIP5, the main conclusions are listed below.

1. Evaluations of downscaled CMIP5 models over China show that CMIP5 projections from NEX-GDDP could better reproduce the monthly temperature and precipitation, but they have systematic errors, which are highly dependent on the region and climate model.
2. In general, CMIP5 is much better in reproducing the monthly minimum temperature than the monthly maximum temperature. The accuracy of temperature is noticeably higher than that of precipitation in CMIP5 models. The ability of MME to reconstruct spatial–temporal information (temperature, precipitation) is better than any single CMIP5 model.

In terms of drought projection and the potential risk for crop production, the main conclusions are summarized below.

1. Taken as a whole, SPEI_3 is projected to significantly decrease across China (−0.15/decade under RCP4.5; −0.14/decade under RCP8.5). SPEI_3 reveals a spatial pattern of drier conditions in high-latitude and high-altitude regions (including NWC, NEC, NC, and the Tibetan Plateau) and wetter conditions in Southern China (including SEC, EC, SC, and SWC), and this pattern will be intensified under RCP8.5.
2. Overall, China is projected to get drier in the four seasons. The decreased rate of SPEI_3 is projected to be largest in winter (−0.2/decade and −0.31/decade), then lower in spring (−0.18/decade and −0.27/decade) and autumn (−0.11/decade and −0.14/decade), and the lowest in summer (−0.08/decade and −0.10/decade) under RCP4.5 and RCP8.5, respectively.
3. From the zonal perspective, winter and spring will get drier in high-latitude/-altitude areas, and summer and autumn will get wetter (especially for Southern China).
4. For moderate and severe droughts, drought frequency and duration are projected to decrease, while both drought frequency and duration will increase for extreme drought. Furthermore, the values of frequency and duration for extreme drought are higher in high-latitude and elevated regions than those in southern regions.

5.    There is a significant increase in the percentage of potential crop production affected by drought over time, where the values will increase up to 36% (RCP4.5) and 47% (RCP8.5) at the end of this century. The ratio caused by extreme and severe drought areas are projected to show an increasing trend with larger magnitudes under RCP8.5.

Moreover, the results of this study will potentially help the government to establish drought management plans by integrating drought projections (e.g., spatiotemporal trends and severity from this study) into macroscale policies to upgrade current practices. This support includes providing farmers with climate risk management early warning and prevention information for the projected climate change induced trends and spatiotemporal variability of future droughts.

Given the main limitations of the current datasets and methods, future research should strive to generate even higher spatial resolution climate projection data, thus enabling drought impact on crop yields to be investigated on practical farming field scales. On the other hand, the adopted SPEI-HS method could potentially provide a more accurate drought risk assessment by combining higher-resolution climate projection data with dynamic land cover use forecasting models. Furthermore, this study suggests that China needs to continue to expand its policy to support farmers in fighting drought, but it may also be worth examining the costs and benefits of direct government policy support on the microeconomic scale.

**Author Contributions:** All authors contributed to this manuscript. Z.L., J.C., L.Y. and Y.H. designed the methods and collected the data; X.G., C.Z. and Y.Y. analyzed the data, and all interpreted the results.

**Funding:** This research was funded by the National Natural Science Foundation of China under grant No. 71461010701 and grant No. 41701385; and supported by the National Key Research and Development Program in China under grant No. 2017YFA0603600. This research was supported in part by Volvo Group in a research project of the Research Center for Green Economy and Sustainable Development, Tsinghua University, and Tsinghua China Data Center and China Siyuan Foundation for Poverty Alleviation under the project "How can public policy guide China's environmental regulation and poverty alleviation".

**Acknowledgments:** Reviewer and editor comments are highly appreciated.

**Conflicts of Interest:** The authors declare no conflicts of interest.

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
