# Peer review of "Drought Trend Analysis Based on the Standardized Precipitation–Evapotranspiration Index Using NASA’s Earth Exchange Global Daily Downscaled Projections, High Spatial Resolution Coupled Model Intercomparison Project Phase 5 Projections, and Assessment of Potential Impacts on China’s Crop Yield in the 21st Century"

_water, doi:10.3390/w11122455_

Round 1

Author Response

Comments and Suggestions for Authors

The manuscript "Drought trends analysis based on SPEI using NEXGDDP high-spatial resolution CMIP5 projection and Assessment of its potential impact on crop yield over China in the 21st century" by Guo et al. presents a study of projected drought changes based on SPEI over 8 sub – regions of China under different RCP scenarios in the 21stcentury by using high-resolution climate data of daily precipitation, maximum daily temperature, and minimum daily temperature from NEX-GDDP project from 10 different CMIP5 models. The manuscript is well organized and fits the scope of Water journal. However, I consider that major changes must be done in order to improve the reliability of the results. Read below my questions and suggestions.

Thanks for your positive and constructive comments. Our response listed below

Introduction

- From lines 1 to 4 you mention the impact of global warming on the frequency of extreme weather events. Also mention that the frequency and strength of drought, will continue increasing. First, in my opinion drought is not an extreme weather event, you neither say that, but perhaps you should refer to extreme hydrometerological, hydroclimatological events instead of extreme weather events. You must add more explanations in this part. If global warming increase the amount of atmospheric water vapor, and consequently the precipitation, why the frequency of droughts is expected to increase. Is just to clarify these topics.

Response: Yes, we would also agree. The introduction was written andthe text was revised as “Under climate change and warming temperature, the water cycle (including precipitation, evaporation, soil moisture and runoff etc.) is consequently accelerating, increasingly causing extreme hydrometeorological or hydroclimatic events (such as floods and droughts) across the world [1-4]. The special report (IPCC SREX, 2012) also points out that more frequent and severe floods and droughts will likely continue increasing in coming decades. Particularly, droughts have immensely adverse impact on water resources and crop yields.“

- Please, define RCP in the text.

Response: Thanks for paying attention to the detail. we added the definition of RCP (Representative Concentration Pathways.) in the text (footnot1 in page 1).

- Please, add more references for this sentence: In general, by using Coupled Model Inter-comparison Project Phase (CMIP) at spatial resolution of 0.5°~2°, these studies have suggested that both frequency and severity of drought would continuously increase with global warming.

Response: Sorry that we were not clear in our expression. We added references and rewrote the sentence.

“Moreover, future change in drought has been analyzed under different Representative Concentration Pathways (RCP) scenarios by using Coupled Model Inter-comparison ProjectPhase (CMIP); and the reliability of CMIP5 to project the spatial-temporal variation of drought in China have been assessed and validated by several studies [27-30], but in general, at coarse spatial resolution of 0.5°~2°. Although these studies have suggested that both frequency and severity of drought would continuously increase with global warming; however, distinctive contradictions exist among these studies [32-36].”

- In this sentence: However, for the spatial change of dry and wet conditions in terms trend and magnitude over China, distinctions exist among these studies. Among which studies? Are these studies 17-20?

Response: Again thank you for paying attention to the details, the text revised as above.

- It is not usual to describe the dataset utilized in the section 2. Study area and datasets…. Before the Methods!

Response: Thanks a lot. The structure was corrected: 1. Introduction/ 2. Datasets: CMIP5 simulations and forcing data/ 3. General method and Study area/ 4. Simulation Results and discussion / 5. Conclusions.

- It is well known, but explain or add a formula for the RMSE.

Response: We added a formula and brief explanation of RMSE in footnote 4 (page 5), and revised the text.

“And the accuracy of the dataset has been evaluated and the previous researches show that meteorological parameters from this dataset have less root-mean-square errors (RMSE[1]and higher correlation coefficients (CC) with ground measurements over the other dataset.”

“, we assume that we have n samples, and  represent model error. RMSE has been used as a standard statistical metric to measure model performance [80].”

- For this sentence you could add more references: Compared with Thornthwaite method, Hargreaves-Samani equation (HS) equation shows advantages in both accounting for the atmospheric information and capturing part of land surface properties (soil moisture, land surface) [53].

Response: Thanks for the suggestions. we added more references.

Almorox, J., V. H. Quej, and P. Martí (2015), Global performance ranking of temperature-based approaches for evapotranspiration estimation considering Köppen climate classes, Journal of Hydrology, 528, 514-522, doi:https://doi.org/10.1016/j.jhydrol.2015.06.057. Hargreaves, G. H. (1989), Accuracy of estimated reference crop evapotranspiration, Journal of Irrigation & Drainage Engineering, 115(6), 1000-1007. Hargreaves, G. H., and R. G. Allen (2003), History and evaluation of Hargreaves evapotranspiration equation. J Irrig Drain Eng ASCE, Journal of Irrigation & Drainage Engineering, 129(1), 53-63. Hargreaves, G. H., and Z. A. Samani (1985), Reference Crop Evapotranspiration from Temperature, 1(2). doi:10.3972/westdc.002.2014.db. Rhee, J., and J. Cho (2015), Future Changes in Drought Characteristics: Regional Analysis for South Korea under CMIP5 Projections, Journal of Hydrometeorology, 17(1), 151016095329004.

- You mention the Hargreaves-Samani equation, and it advantages respect the Thornthwaite; so you can add the HS equation. In addition, FAO recommend to utilize the HS equation in the absence of enough meteorological data to compute Eto through the Peaman-Montie equation, please mention this!

Response: Thanks. This is a very valid point. Now we have added the HS equation in footnote 2 and the FAO recommendation:

Hargreaves and Samani equation (Hargreaves and Samani, 1985): . Where  is evapotranspiration in mm/dekad;  is the empirical coefficient;  are the dekadal mean, maximum and minimum air temperatures in °C;  is the extraterrestrial radiation (MJm-2); d is the number of days within the dekad.

- Page 5. …. and also have been used in SPEI calculation in previous works [50]. But you just mention one reference!

Response: Thanks for paying attention to the details. We have added more references and revised this part.

“Standardized Precipitation Evapotranspiration Index (SPEI), carried out by Vicente-Serrano et al. in 2010 [52], is a multiscales drought index based on climatic data. It can describe water deficit effectively with multiple time scales, reflecting the lag relation between different water resources, precipitation, and evapotranspiration [14]. Three-month SPEI was employed in this study to characterize short duration droughts [53]. More details about SPEI can be obtained in its website (http://sac.csic.es/spei/index.html) and previous papers [52,54-56].

SPEI was preferred compared to the standardized precipitation index SPI [57] since the former considers the effect of the atmospheric circumstances to the water balance by introducing evapotranspiration. Under the global warming, the SPEI not only retains the multi-scale characteristics of droughts [52] in SPI, but also helps account for the impact of temperature. Therefore, SPEI has been nominated as a substitute of SPI as well as Palmer drought severity index (Palmer, 1965) to evaluate climatic water imbalance and drought [58-61]. “

- Page 5. Three-month timescale was chosen for SPEI in drought projection, whichcould be considered as agriculture drought. Please, add a reference for this !!!

Response:  Yes, you are right, and this paper also focus on drought impact on crop yields. We have added further references to emphasize that 3-month SPEI could effectively evaluate the effect of drought on crop yield.

The results from Bo et al (2015) indicated that detrended yields estimated reflected the real situation in history and could be used to study the relationships between yield variations and climate elements, andhas shown that three-month timescale, especially from June to August, was highly correlated with detrended yield, which can effectively evaluate the effect of drought on crop yield.

- Can the authors explain more clearly how they calculate mean error ME for temperature and BIAS for precipitation?

Response: Sorry we didn’t clarify it clear. The ME/BIAS of each region equals the average ME/BIAS of corresponding pixels.

- The authors validate precipitation, maximum temperature, and minimumtemperature from NEX-GDDP 10 CMIP5 models calculating correlationcoefficient (CC) and mean error (ME); however, in my opinion, you should also assess the accuracy of the models in reproducing the SPEI during the historical period (1979-2000) to compare with the SPEI computed with data from ITPCAS. The question is: are able the models to reproduce drought conditions (episodes, duration, frequency) during the reference period? And you don’t assess this but at the same time you highlight the fact that modelling data are already BIAScorrected.

Response:  Thanks. This is a very valid point.The time set 1979-2000 is not long enough to reproduce drought, the ITPCAS dataset in this research was used to evaluate the accuracy of NEX-GDDP. The 1979-2000 is just the evaluation period, rather than reference period, to assess the climate model data accuracy. And the ITPCAS data is only available during that period. The downscaled method used in NEX-GDDP Dataset and widely used in climate change simulation downscale is Bias-Correction Spatial Disaggregation (BCSD) method [41-43]. In BCSD, the global observed data used as reference is from Global Meteorological Forcing Dataset for Land Surface Modeling, which was produced by the Terrestrial Hydrology Research Group at Princeton University [44]. More detail information could found in the paper of Thrasher et al. (2012) [35].

- In the section 4.4. you described the drought duration and frequency! Define the way you utilized to obtain both in the methodology.

Response: Sorry we didn’t clarify it clear.

For each ten years, the drought frequency of each region equals the average frequency of the corresponding pixels; For each ten years, the drought duration of each region equals the average duration of the corresponding pixels.

- It is confusing to establish a reference period and later investigate trends for thewhole 1950-2100! Why before 1979?

Response: Sorry we didn’t clarify it clear. Actually the 1979-2000 is just the evaluation period, rather than reference period, to assess the climate model data accuracy. And the ITPCAS data is only available during that period.

Minor comments:

Abstract, Line 2. “We quantified the drought projections from very high-resolutionclimate data and and further…” delete one and.

Response: We apologize for the typo errors. This has been corrected now.

Section 3.1 line 7. D (=P-PET), please put in correct form D = (P-PET).

Response: we corrected this.

Section 3.1 Table 2. Put the reference from where you used criterion in the table 2.

Response: The reference has been added.

Section 4.1, page 7 line 1. (“As presented in Figure 3, using GCMs to predict temperature is better than its precipitation of precipitation (Fig. 2)”), please write this more clear

Response: Sorry we didn’t clarify it clear. Now the text is revised as “the performance of using GCMs to predict temperature is better than using GCMs to predict precipitation”.

Page 8. “The change of SEPI on regional scale is summarized in Table 3”. Pleasechange SEPI in SPEI

Response: Thanks. We corrected this.

Reviewer 2 Report

Attached

Round 2

Reviewer 1 Report

The article "Drought trends analysis based on SPEI using NEX-GDDP high-spatial resolution CMIP5 projection and Assessment of its potential impact on crop yield over China in the 21st century" can be accepted in current form.

Author Response

Thanks a lot for your positive comments.

Reviewer 2 Report

Although there has been a substantial improvement of the revised document and a response has been provided to all comments previously raised, there are still some important aspects which should have been already addressed in this submission:

It is very important that the English language is carefully revised throughout the manuscript (there are many sentences which are very difficult to understand).

It is important that the responses that the Authors provided to the questions previously raised by the Reviewer are included in the manuscript (or most of them). This information can help the future reader of the manuscript (this is, questions were not raised only to provide the Reviewer a response but mostly thinking in the future reader). For example:

Why the diagram showing the methodology approach has not been included in the manuscript? Why the mathematical formulae has not been added? etc

References to Tables and Images needs to be carefully revised throughout the manuscript (for example, erroneous reference to Table 2)

Why are there comments within the submitted revised manuscript (and some of them in different language)? Was the document submitted a final document or still draft version?

Author Response

This manuscript is a resubmission of an earlier submission. The following is a list of the peer review reports and author responses from that submission.